# Programmable broadband optical field spectral shaping with megahertz resolution using a simple frequency shifting loop

Côme Schnébelin[1], José Azaña[2] & Hugues Guillet de Chatellus[1]*

Controlling the temporal and spectral properties of light is crucial for many applications. Current state-of-the-art techniques for shaping the time- and/or frequency-domain field of an optical waveform are based on amplitude and phase linear spectral filtering of a broadband laser pulse, e.g., using a programmable pulse shaper. A well-known fundamental constraint of these techniques is that they can be hardly scaled to offer a frequency resolution better than a few GHz. Here, we report an approach for user-defined optical field spectral shaping using a simple scheme based on a frequency shifting optical loop. The proposed scheme uses a single monochromatic (CW) laser, standard fiber-optics components and low-frequency electronics. This technique enables efficient synthesis of hundreds of optical spectral components, controlled both in phase and in amplitude, with a reconfigurable spectral resolution from a few MHz to several tens of MHz. The technique is applied to direct generation of arbitrary radio-frequency waveforms with time durations exceeding 100 ns and a detection-limited frequency bandwidth above 25 GHz.

[1] University Grenoble Alpes, CNRS, LIPhy, 38000 Grenoble, France. [2] INRS-EMT, 800 de la Gauchetière Ouest, Suite 6900, Montréal, QC H5A-1K6, Canada.
*email: hugues.guilletdechatellus@univ-grenoble-alpes.fr

The customized control of the time and/or frequency-domain field of an optical waveform, usually referred to as optical arbitrary waveform generation (OAWG), is fundamental for many important applications, e.g., for the coherent control of quantum processes[1–5], implementation of critical functionalities in telecommunication systems[6] and microwave photonics applications[7–12]. Key performance parameters in an OAWG system are the overall range of frequencies that can be synthesized (bandwidth) and the spectral resolution with which one can manipulate the optical spectrum. The former defines the time resolution of the generated optical waveforms and the later determines their maximum time duration. The ratio of these two parameters (bandwidth/frequency resolution) is known as the time-bandwidth product (TBP) and often used as a main performance specification to evaluate the complexity of the waveforms that can be generated with a given system.

Among the numerous approaches proposed to date for OAWG[13,14], the Fourier synthesis method[7] remains the most popular one, and in fact, programmable pulse-shaping devices based on this principle are commercially available. This method is based on user-defined filtering of the amplitude and phase profiles of a broadband phase-coherent optical spectrum, e.g., a short optical pulse from a mode-locked (ML) laser. This scheme has proven suitable for the synthesis of relatively complex ultra-broadband optical waveforms, with a large time-bandwidth product (TBP > 100). However, the frequency resolution offered by this solution is usually poorer than ~10 GHz, which constrains the duration of the synthesized waveforms to below the sub-nanosecond regime[15]. The combination of high-resolution spatial-domain pulse shapers with high repetition rate ML lasers has enabled spectral line-by-line shaping of the corresponding optical frequency comb (periodic set of discrete frequency lines that defines the spectrum of a repetitive optical pulse train) with a resolution of only a few GHz[16,17]. However, this has been achieved at the expense of large insertion losses, lower TBP, and other technical challenges. Other approaches using the Fourier synthesis concept have also been proposed, including solutions based on acousto-optic programmable dispersive filters[18], space-to-time mapping[19], or through a coherent combination of mutually phase-locked lasers[20]. However, these approaches may still require the use of a broadband ML laser[18,19], and most importantly, they are very challenging to scale for the synthesis of complex waveforms (with a large TBP).

Recently, to overcome the difficulties associated to the use of a ML laser, OAWG solutions have been developed that employ spectral shaping of optical frequency combs directly generated from a single continuous wave (CW) laser. The comb can be generated by non-linear interactions in a micro-resonator[21], or by electro-optic modulation[22]. In these solutions, the comb frequency spacing can be adjusted to match the shaper's resolution, enabling efficient line-by-line spectral shaping[23–27]. However, the number of spectral lines in these schemes, i.e., the TBP, is still limited to ~100[24]. Moreover, these methods still suffer from the frequency resolution limitations of Fourier filtering techniques, thus being unsuited for generation of arbitrary optical waveforms with durations beyond the sub-nanosecond regime.

A different approach for optical field spectral shaping is based on the direct generation of the target complex spectrum by temporally modulating a CW laser[28]. The frequency resolution of this technique is only constrained by the spectral linewidth of the CW laser. However, increasing the number of frequency components remains a technological challenge, and the performance (e.g., frequency bandwidth) of the method is ultimately limited by the electronic arbitrary waveform generation system (AWG) driving the modulators[29]. Extensions of this method to enable broader bandwidth operations have been demonstrated through joint spectral filtering and line-by-line temporal modulation, but these systems are inherently very complex and still require the use of a broadband ML laser[30,31].

In this paper, we demonstrate a concept for reconfigurable OAWG, where an arbitrary broadband optical spectrum, with a user-defined amplitude and phase spectral profile, is synthesized directly from a CW laser seeding a recirculating frequency shifting loop (FSL). Beforehand, the CW laser is temporally modulated, in amplitude and phase, by a low-bandwidth (typically < 100 MHz) input electrical signal. After multiple rounds through the FSL, the instantaneous optical spectrum of the light is re-shaped proportionally to the temporal shape of the modulation input signal (in amplitude and phase), i.e., a time-to-frequency mapping process is achieved. This simple scheme enables the synthesis of arbitrary optical spectra, with hundreds of frequency components simultaneously controlled both in amplitude and in phase. Most importantly, this is achieved with a spectral resolution that is reconfigurable from a few MHz to several tens of MHz, and over an optical bandwidth (BW) that can potentially exceed the hundreds of GHz range. To showcase the unique capabilities offered by this new concept for OAWG, we report here direct generation of arbitrary broadband RF waveforms, both baseband and on an RF carrier, with a TBP exceeding 300, with a frequency resolution as narrow as ~9.5 MHz (time duration > 100 ns) or a spectral BW as large as 25 GHz (frequency resolution: ~85 MHz), limited by the available detection bandwidth. Additionally, the carrier frequency of the generated RF waveforms can be arbitrarily controlled, from DC to a few tens of GHz. The unprecedented flexibility and performance of this simple and low-cost platform should fulfill the requirements for numerous applications in physics, telecommunications, photonics, and microwave engineering.

## Results

**Theoretical description**. The OAWG concept described here is based on a recirculating frequency shifting loop (FSL) seeded by a narrow-linewidth monochromatic (CW) laser[32–34]. Recall that a FSL comprises an acousto-optic frequency shifter (AOFS) and an amplifier with moderate gain, to compensate for the losses in the loop. A tunable optical bandpass filter (TBPF) is also inserted in the FSL, both to control the spectral bandwidth and to limit the amplified spontaneous emission (ASE) generated by the amplifier. We define $f_s$ as the frequency shift per round-trip induced by the AOFS, and $\tau_c$ as the propagation time in the loop.

An optical coupler enables to seed the FSL with a CW laser (frequency: $f_0$), temporally modulated in amplitude and phase by means of an acousto-optic modulator (AOM). We define $N$ as the ratio between the optical bandwidth of the TBPF, and $f_s$: $N$ corresponds to the maximum number of round-trips of the input light in the loop. The output optical signal is obtained by extracting a fraction of the light from the loop by means of a second optical coupler (Fig. 1). For characterization purposes, the measurement of the instantaneous optical spectrum of the output waveforms is carried out by recombining the output signal with a fraction of the seed laser on a photodiode (see Methods). The time-domain heterodyne beating is acquired and processed (e.g., Fourier transformed) off-line by means of a digital sampling oscilloscope (DSO). When the reference beam is blocked, the detection system provides a direct measurement of the temporal intensity waveform at the FSL output.

Let us assume that the seed laser is modulated by an electrical signal $e_{in}(t)$ with a duration $\tau > \tau_c$, and with a bandwidth narrower than $1/\tau_c$. Said other way, the input modulation signal is approximately constant at the time scale of one round-trip in the loop. We define $E_{in}(t) = e_{in}(t)e^{i2\pi f_0 t}$ as the optical field injected in

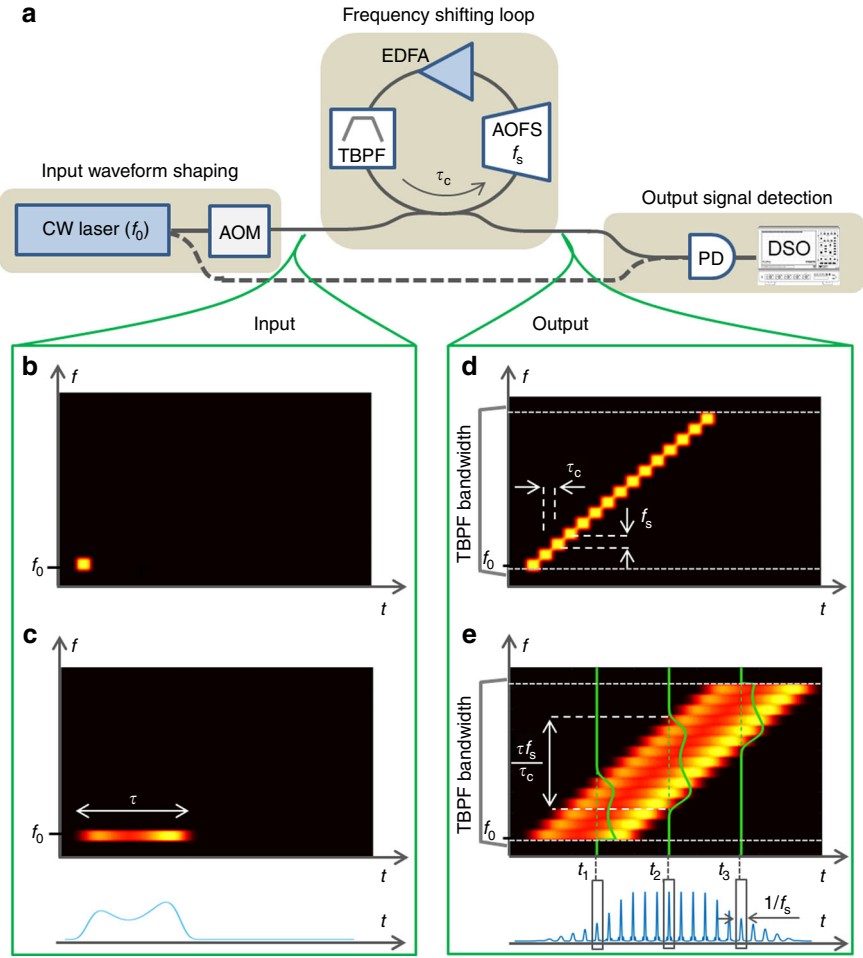

**Fig. 1** Principle of arbitrary optical field spectrum synthesis and control. **a** experimental setup. The input waveform shaping unit comprises a seed CW laser (frequency: $f_0$) and an acousto-optic modulator (AOM). The FSL (travel time: $\tau_c$) comprises an acousto-optic frequency shifter (AOFS) driven at $f_s$, a tunable bandpass filter (TBPF), and an optical amplifier (EDFA). The output signal is detected by a fast photodetector (PD) and numerically processed with a fast digital sampling oscilloscope (DSO). A fraction of the CW seed laser can be used as a reference, for heterodyning with the optical output of the FSL (dashed line). An optional time-gating device can be inserted to extract an individual waveform from the output train. **b–e** intuitive explanation of the technique in the time-frequency plane, with a short (**b**, **d**: duration ~$\tau_c$) and a long (**c**, **e** duration $\tau > \tau_c$) injection signal. The signal at the FSL output consists of a set of $N$ (here $N=15$) replicas of the input signal, simultaneously shifted in time and in frequency, by $\tau_c$ and $f_s$, respectively, where $N$ is determined by the TBPF bandwidth ($<N \times f_s$). When the input signal has not completely entered the FSL (e.g., at $t_1$) or when the leading edge of the frequency-shifted input signal has been filtered out by the TBPF (at $t_3$), the instantaneous output frequency spectrum, i.e., the spectrum measured by time-gating the output signal over a duration of $1/f_s$ (in green), maps only a truncated version of the input modulation signal (in light blue). For intermediate times (e.g., at $t_2$), the instantaneous optical spectrum maps the whole input modulation signal, and it is shifted by $f_s$ every $\tau_c$. In this latter case, the intensity of the output signal consists of a set of identical pulsed waveforms spaced by $1/f_s$

the FSL (analytical representation), where $e_{in}(t)$ is the field temporal complex envelope. The optical field circulating in the loop is constructed from the coherent addition of replicas of the input field, shifted along the time and the frequency domains by $\tau_c$ and $f_s$, respectively. In the case where the power of the EDFA is adjusted so as to compensate for the losses of the loop, neglecting the amplified spontaneous emission (ASE) emitted by the amplifier, the output optical field can be expressed as follows[35]:

$$E_{out}(t) = \gamma \sum_{n=0}^{N} E_{in}(t - n\tau_c) e^{i2\pi n f_s t} e^{i\pi n(n+1) f_s \tau_c} \quad (1)$$

where $\gamma$ is a complex term that depends on the transmission coefficient of the couplers, and the gain and losses in the FSL. The simultaneous shifts along the temporal and spectral domains induce a quadratic phase term: $e^{i\pi n(n+1) f_s \tau_c}$. Notice that this quadratic phase term induces a temporal Talbot effect when $f_s \tau_c$ is equal to a ratio of two mutually prime integers[33]. It also enables

the generation of reconfigurable optical chirped waveforms when $f_s \tau_c$ is close, but not equal, to an integer value[36]. In the following, $f_s$ is chosen such that $f_s \tau_c$ is an integer; under this condition, this quadratic phase term reduces to unity, and the output optical field re-writes:

$$E_{out}(t) = \gamma \sum_{n=0}^{N} e_{in}(t - n\tau_c) e^{i2\pi(f_0 + n f_s)t} e^{-i2\pi n f_0 \tau_c}. \quad (2)$$

In the case when no modulation is applied to the seed laser ($e_{in}$ is constant), the output optical field $E_{out}(t)$ consists of a comb of phase-locked optical frequencies starting at $f_0$ and separated by $f_s$, over a total bandwidth $N f_s$. This set of optical frequencies exhibits a linear phase profile, $e^{-i2\pi n f_0 \tau_c}$, simply corresponding to an overall time delay of the resulting waveform. In particular, in the temporal domain, the output signal is a train of Fourier transform-limited pulses, repeating at a rate of $f_s$. When the modulation signal is applied to the seed laser, the comb lines are

slightly broadened by the modulation function $e_{in}(t)$. As illustrated in Fig. 1 and according to the mathematical description in Eq. (2), at each round-trip, the input modulation waveform is simultaneously shifted in time and in frequency by $\tau_c$ and $f_s$, respectively. After a number $R$ of round-trips, the signal at the output of the FSL is composed by the sum of $R$ time- and frequency-shifted replicas of the input signal. As such, the output waveform at any given instant of time can be described as a superposition of $R$ consecutive temporal segments of the input signal, each with a duration of $\tau_c$, and each shifted in frequency with respect to each other by $f_s$. When the overall number of round-trips is such that the entire input waveform is already within the FSL cavity, that is when $R\tau_c > \tau$ (total time duration of the input waveform), the time segments of the input signal that are summed up to form the corresponding instantaneous output waveform will cover the full input signal duration. As each of these time segments is shorter than the fastest time feature of the input waveform (recall that the signal bandwidth is narrower than $1/\tau_c$) and these time segments are also consecutively shifted along the frequency domain, the described process will effectively induce a mapping of the entire input temporal waveform into the frequency spectrum at the FSL output, see illustrations in Fig. 1 for the case e (long input signal) at the observation time $t_2$.

The intuitive picture of the operation principle shown in Fig. 1 highlights the time-to-frequency mapping process, as well as the main design trade-offs in the system. In particular, the scaling law of the implemented time-to-frequency mapping process is simply defined by the ratio $f_s/\tau_c$. The spectral bandwidth of the output signal is set to $Nf_s$ by the TBPF; as such, the parameter $N$ determines the maximum number of round-trips that a signal can go through the FSL cavity. This implies that for a full time-to-frequency mapping, the duration of the input modulation signal must be set to satisfy the following condition: $\tau < N\tau_c$. Moreover, the mapping of the entire temporal input modulation signal to the instantaneous output spectrum effectively holds after the input signal has completely entered the FSL, i.e., after an $R$ ($<N$) number of round-trips such that $R\tau_c > \tau$, and until the leading edge of the input signal is filtered out by the TBPF (Fig. 1), i.e., until the signal undergoes $N$ round-trips through the FSL cavity.

A rigorous mathematical derivation of the concept is given in the Methods section. Briefly, from Eq. (2), it can be shown that the resulting time-domain signal at the output of the FSL consists of a train of consecutive waveforms separated by $1/f_s$, here labelled by m (=1, 2, 3 …) (Fig. 1). The baseband Fourier spectrum of the optical waveform (i.e., Fourier spectrum of the corresponding field temporal complex envelope) emitted at time $m/f_s$, extending over a duration of $1/f_s$, can be written as

$$\tilde{e}_{out}(f, m) \propto e_{in}\left(\frac{m - f\tau_c}{f_s}\right) \qquad (3)$$

for $0 < f < Nf_s$, where $f$ is the optical baseband frequency (i.e., with respect to $f_0$). In practice, the optical waveform emitted at time $m/f_s$ can be extracted from the output pulse train by time gating (e.g., by means of a temporal intensity modulator). The spectral truncation realized by the TBPF implies also that $\tilde{e}_{out}(f, m) = 0$ for $f < 0$ or $f > Nf_s$. Equation (3) highlights the mapping of the input complex temporal modulation signal from the time to the frequency domain. Notice also that the central frequency of the baseband output spectrum evolves linearly with $m$, i.e., with the evaluation time $m/f_s$. This feature can be easily understood by considering that the waveform emitted at $m/f_s + \tau_c$ has gone through one more round-trip (i.e., one more frequency shift) than the one emitted at $m/f_s$.

The spectral bandwidth of the output waveform is equal to $f_s/\tau_c \times \tau$, i.e., proportional to $\tau$, the duration of the input modulation

signal. Recall that the bandwidth of the input modulation signal is assumed to be narrower than $1/\tau_c$, which translates into a temporal resolution of the modulation waveform longer than $\sim\tau_c$. This corresponds with a minimum spectral resolution of the synthesized output optical signal that is approximately equal to the frequency shift induced by the AOFS, $\sim f_s$. As a result, the maximum number of spectral components in the synthesized output waveform (i.e., the maximum time-bandwidth product, or TBP) is given by $\tau/\tau_c$. As discussed above, this ratio is limited by $N$, the maximum number of round-trips that can be achieved in the loop. Experimentally, this parameter is determined by the spectral bandwidth of the in-cavity TBPF, and can reach values above 1000[36]. Notice also that the bandwidth of the output optical waveform ($\sim\tau f_s/\tau_c$) is about $\tau f_s$ times, i.e., orders of magnitude larger than the bandwidth of the input modulation signal ($<1/\tau_c$). This enables generation of broadband optical waveforms (e.g., with a bandwidth up to hundreds of GHz[36]) at the FSL output from low-bandwidth electronic input signals (typ. <10 MHz). Finally, it is also important to recall that in the case where the product $f_s\tau_c$ is not an integer, an additional quadratic component would be imprinted onto the spectral phase of the output optical waveforms, as indicated in Eq. (1), potentially providing an additional degree of control on the generated output spectra[35,36].

**Demonstratio n of arbitrary optical field spectral shaping**. To demonstrate the validity of the proposed concept for high-resolution arbitrary optical field spectral shaping, we first implemented a fiber FSL, with a round-trip propagation time $\tau_c = 105$ ns, which incorporated two AOFS providing a net frequency shift equal to $f_s = 1/\tau_c = 9.482$ MHz (see Methods). The maximum number of round-trips in this first demonstrated system is $N \sim 300$. The loop is injected with a CW laser temporally modulated by an AOM. The latter is driven by a sine wave at $f_m = 80$ MHz, modulated in amplitude and phase by the input RF signal, as detailed in the Methods section. $f_0$ is defined as the carrier frequency of the light wave at the input of the FSL. As predicted, the signal at the output of the FSL consists of a train of waveforms spaced temporally by $1/f_s$ (Fig. 2a). For an accurate characterization of this output optical waveform, the resulting optical spectrum is down-converted to the RF domain by heterodyning the output signal with a fraction of the seed CW laser (Fig. 1)[34]. The resulting light field is sent to a photodetector and the obtained photocurrent is recorded by a 3.5-GHz-bandwidth oscilloscope. A single waveform is numerically extracted from the recorded signal: its Fourier transform (FT) corresponds to the complex-field optical spectrum of the selected waveform down-converted to baseband (i.e., with respect to $f_0$). This setup allows us to measure the Fourier spectrum (amplitude and phase) of the individual optical waveforms at the system output. In the example shown in Fig. 2, the CW laser is modulated in amplitude by a 2-µs long square-like (flat-top) signal (in light blue). The output waveforms after recombination are recorded (dark blue), and the spectra are numerically calculated for each of the individual traces (Fig. 2c). As expected, all waveforms share the same flat-top spectral shape. Notice that the high relative level of the low-frequency components in the experimentally recovered spectra of the output waveforms is due to the relatively low power of the CW laser used in the heterodyne recombination process. The waveforms generated before the whole input signal has entered the FSL (e.g., around the time $t = 0.66$ µs in the shown examples) exhibit a frequency spectrum that maps a truncated version of the input square temporal modulation signal, i.e., a narrower flat-top. Consistently with Eq. (3), when the whole input signal has entered the FSL, the width of the spectral shape remains

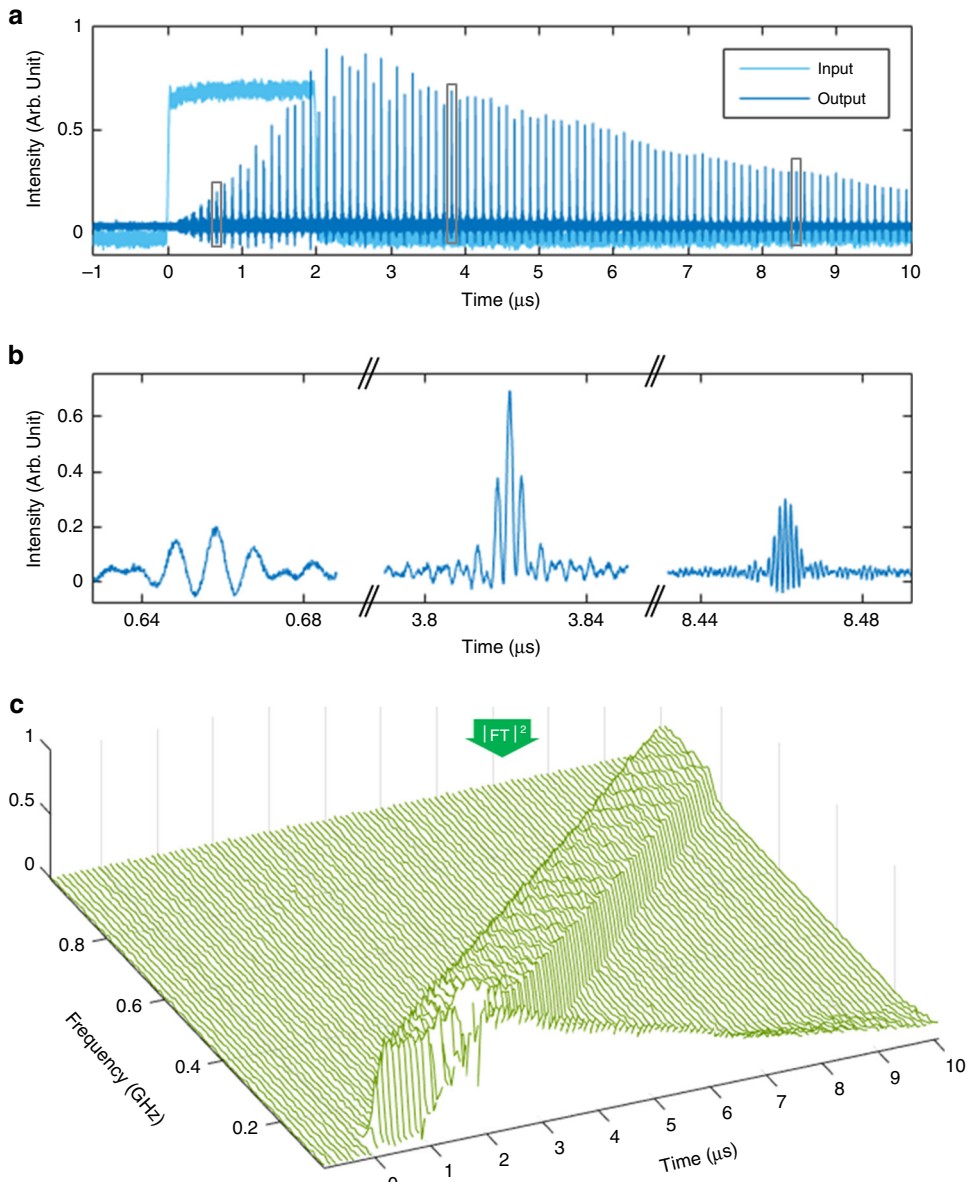

**Fig. 2** Demonstration of time-to-frequency mapping in FSLs. **a** in dark blue, temporal evolution of the output waveforms after heterodyning with the CW laser. The intensity of the input modulation signal (here, a square-like or flat-top signal) is plotted in light blue. **b** selected individual output waveforms after heterodyning with the CW laser. **c** instantaneous power spectrum of the output waveforms after heterodyning (in green). The output individual waveforms are numerically extracted from the recorded time trace, and Fourier transformed. The frequency axis represents the baseband frequency (i.e., the relative optical frequency with respect to $f_O$). Notice that the input signal was imprinted onto the CW laser by means of an AOM driven at 80 MHz, which explains the shift of the lowest output frequencies from DC to 80 MHz. For any time lower than 2 µs, only the leading edge of the modulation signal has entered the FSL, resulting in a truncation of the (flat-top) optical output spectrum. At later times, the instantaneous spectrum shifts by $1/\tau_c$ (=9.482 MHz) every $1/f_s$ (=105.5 ns)

unchanged and equal to ~180 ± 10 MHz, in good agreement with the value predicted by the time-to-frequency scaling law $f_s\tau/\tau_c =$ 181 MHz. Moreover, as predicted, the absolute central frequency of consecutive waveforms increases as the number of round-trips in the FSL is also increased, with a frequency increment of $f_s$ every round-trip ($\tau_c$), equivalent to a frequency increase of $1/\tau_c$ between consecutive individual waveforms (separated by $1/f_s$).

In a second set of experiments, we demonstrated the versatility of the concept for generation of arbitrary optical spectra with user-defined amplitude and phase profiles. Plots in Fig. 3a–d are obtained when the FSL is set with the design conditions defined above. In this case, as expected, arbitrary optical spectra can be synthesized with a frequency resolution equal to $f_s$ (~9.5 MHz).

Plot a shows the instantaneous spectrum (measured in a temporal window of duration $1/f_s$) in the case of an un-modulated seed laser. In this case, the TBPF has been set such that about $N =$ 1000 spectral components are generated over a bandwidth exceeding 10 GHz. Therefore, theoretically the maximum TBP of the technique exceeds 1000. However, in practice, due to the imperfect flatness of the spectrum, the maximum TBP directly achievable (i.e., without compensation of the spectral magnitude) is about ~300. In Fig. 3e, the FSL is now set in a different configuration, where $f_s = 9/\tau_c = 84.91$ MHz, as described in Methods. In this case, the system enables the synthesis of arbitrary optical spectra with a relatively poorer frequency resolution (equal to $f_s$) but over a broader bandwidth, about

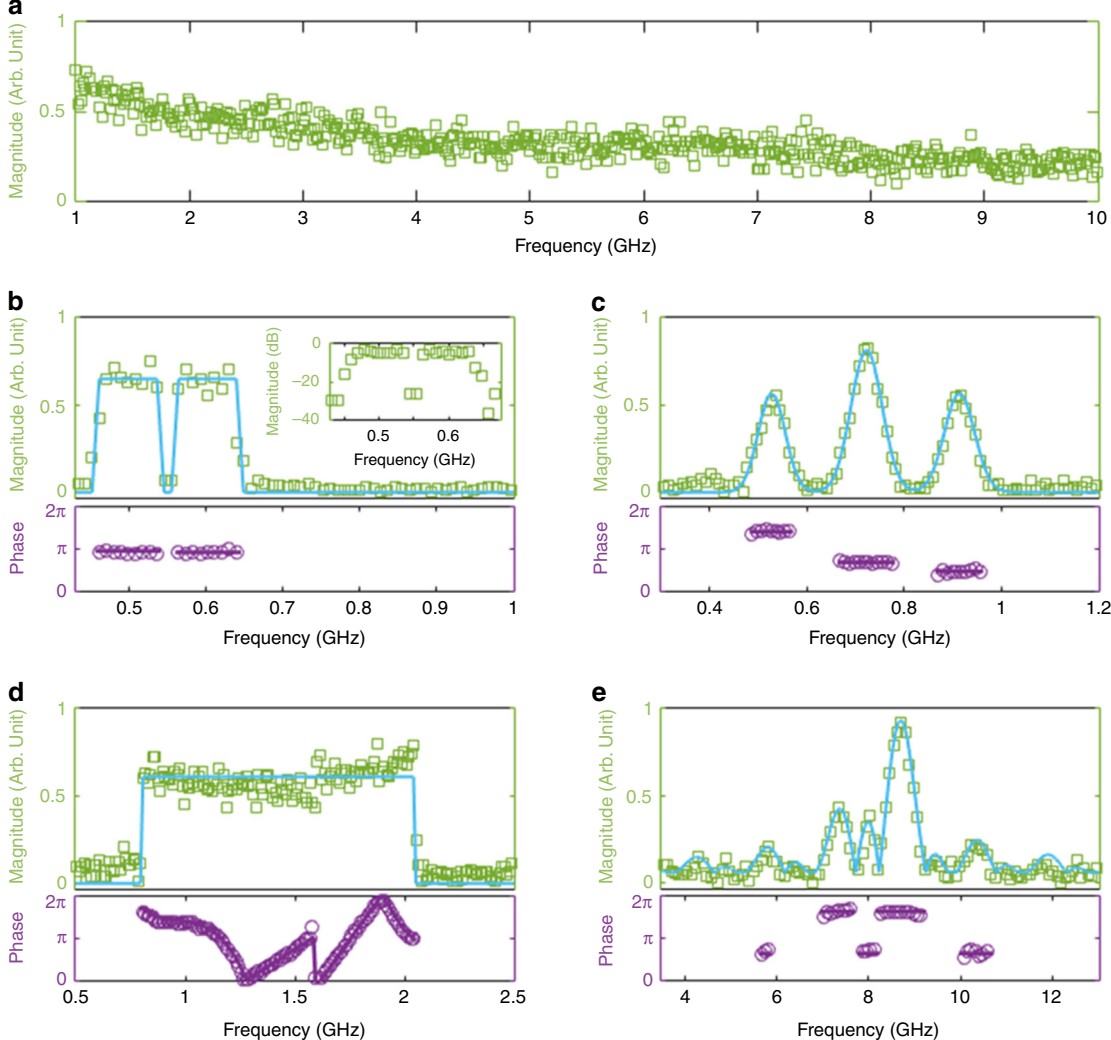

**Fig. 3** Optical spectra with arbitrary magnitude and phase profiles. Experimental data are represented by symbols to show the spectral resolution of the OAWG (green squares: amplitude, purple circles: phase); the light blue (purple) solid line is the amplitude (phase) signal applied to the modulator and mapped into the spectral domain using the theoretical time-to-frequency scaling law (Eq. (3)). The frequency represented along the horizontal axis is the optical frequency relative to the frequency of the seed laser ($f_0$). **a** optical spectrum generated at the output of the FSLs seeded by an un-modulated CW laser. About 1000 spectral components separated by $f_s = 9.482$ MHz are generated. **b** flat-top amplitude with suppressed central frequency (20 MHz spectral bandwidth, 20 dB rejection), and constant phase. **c** combination of three Gaussian amplitudes with distinct uniform phases. **d** flat-top amplitude and composite spectral phase, including cubic and linear profiles. **a**, **b**, **c**, **d** have been obtained with $f_s = 9.482$ MHz. **e** example of broadband arbitrary complex-field optical spectrum synthesis (with $f_s = 84.91$ MHz). For plots **b**, **c**, **d**, **e**, the values of the RMS error between the generated (green) and the designed waveforms (light blue) are, respectively, equal to 0.10, 0.033, 0.080, 0.034 (in arb. units) for the magnitude, and 0.084, 0.090, 0.18, and 0.18 (in radians) for the phase. These values were calculated over the whole displayed frequency range, except in **b**, where they were calculated between 0.43 GHz and 0.7 GHz

nine times larger than in the high-resolution scheme. In all cases, there is an excellent agreement between the experimentally recovered amplitude and phase spectral shapes of the synthesized optical waveforms and those theoretically expected according to the amplitude and phase temporal modulation functions imprinted on the input CW laser.

**Baseband RF-AWG.** The possibility of optical field spectral shaping with a MHz resolution makes this technique particularly attractive for direct application to radio-frequency AWG (RF-AWG). The proposed concept enables generation of baseband RF arbitrary waveforms in a very straightforward fashion, namely, by simple direct photo-detection of the temporal intensity waveform at the output of the FSL (i.e., without heterodyning with the CW laser). In this case, according to the calculations given in

Methods, when the whole input signal has entered the FSL, the photocurrent produced by the waveform $m$ can be simply written as:

$$I_{out}(t, m) \propto \left| \tilde{e}_{in}\left( \frac{f_s}{\tau_c} t \right) \right|^2 \tag{4}$$

where $\tilde{e}_{in}(f)$ is the Fourier transform of the complex-modulation function $e_{in}(t)$ that is imprinted on the input CW light. Equation (4) indicates that the generated RF waveforms are not dependent on the observation time (parameter $m$), or said other way, the target waveform is produced in a repetitive fashion, with a repetition period fixed by the inverse of the frequency shift $f_s$. Additionally, the number of properly shaped waveforms is ultimately limited by the total number of round-trips in the loop ($N$). In this case, the input modulation signal to be applied on the

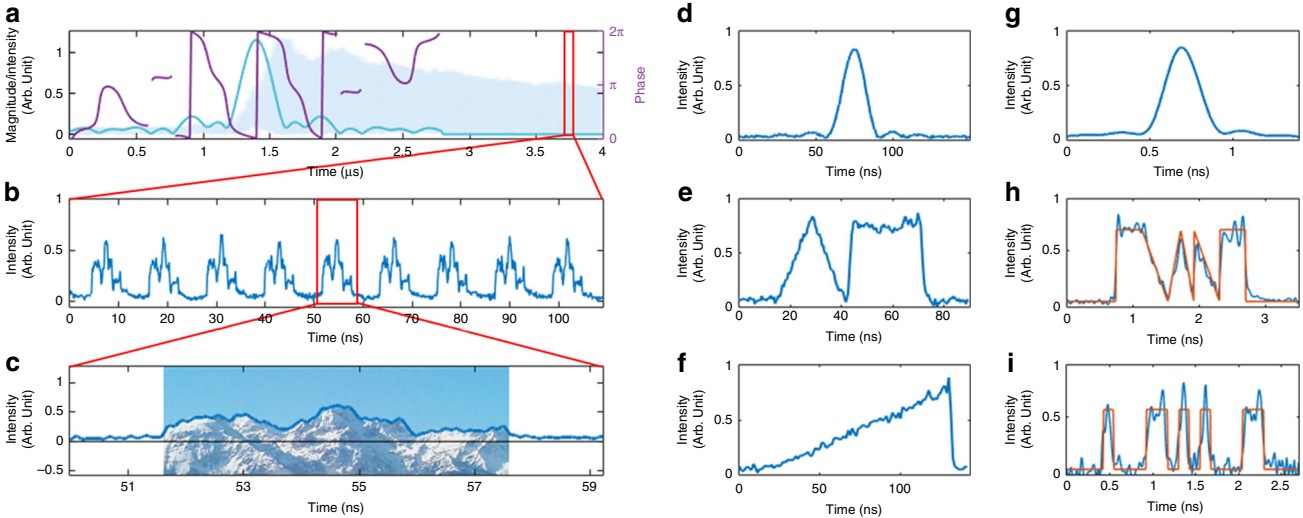

**Fig. 4** Demonstration of baseband RF AWG. **a–c** example of generation of an arbitrary waveform mapping the profile of Belledonne mountain, near Grenoble, France. **a** input modulation signal (light blue: amplitude, purple: phase) and output time trace (blue background). **b** train of nine consecutive emitted waveforms (rep. rate: 84.91 MHz). For clarity, the temporal scale has been shifted by 3.77 μs. **c** comparison of a single output waveform to the desired profile. **d–f** examples of ~100 ns long arbitrary waveforms with various shapes (sinc$^2$, triangular and flat-top, and a linear ramp). The FSL is implemented such as: $f_s = 9.482$ MHz. The maximum signal bandwidth is limited by the 3.5 GHz detection bandwidth. **g–i** broadband baseband AWG (sinc$^2$, broken lines profile, and on-off keying modulation for digital communications at 8 Gbits/s ("0100011010100011" in Binary, corresponding to "46A3" in Hex.). The orange plots correspond to the target waveforms. The FSL is implemented such as $f_s = 84.91$ MHz. The maximum signal bandwidth is 25 GHz, limited by the detection equipment

input CW laser in order to achieve a target waveform is determined by the inverse FT of the square root of the desired output intensity waveform. In Fig. 4a, we plot the input temporal modulation signal (amplitude and phase) enabling the generation of output waveforms mapping an arbitrary signal (here, Belledonne mountain, near Grenoble, see bottom trace). In this case, the system is designed for a frequency shift of $f_s = 84.91$ MHz, and the output bandwidth exceeds 25 GHz, limited again by the detection device. Other examples of arbitrary signals in the same experimental conditions are given in Fig. 4g–i. When $f_s = 9.482$ MHz, the FSL enables direct generation of waveforms with durations exceeding ~100 ns (corresponding to a frequency resolution of $f_s$), and with a frequency bandwidth limited by the detection bandwidth to ~3.5 GHz, see results shown in Fig. 4d–f. The TBP of the synthesized waveforms in this case exceeds 300.

**OAWG and RF-AWG on a carrier**. We now consider the full optical field at the output of the FSL. As detailed in Methods, the complex amplitude of the output optical waveform emitted at a time $m/f_s$, when the entire input signal has entered the FSL, can be written as:

$$E_{\text{out}}(t, m) \propto \tilde{e}_{\text{in}}\left(\frac{f_s}{\tau_c} t\right) e^{i2\pi\left(f_0 + \frac{m}{\tau_c}\right)t}. \quad (5)$$

Recall that $m$ refers to the evaluation time, in steps determined by the inverse of the fundamental frequency shift induced in the FSL, $1/f_s$. Equation (5) implies that the complex envelope (amplitude and phase) of the generated output waveforms is readily set by the FT of the input modulation function, and the carrier frequency evolves linearly with $m$; as described above, the carrier frequency increases by $f_s$ per slot of time corresponding to $\tau_c$, or equivalently, the carrier frequency increases by an amount of $1/\tau_c$ from one waveform to the following one (temporally spaced by $1/f_s$). The maximum value of the carrier frequency corresponds to the waveform having experienced the maximum number of round-trips $N$, and is approximately given by $f_0 + Nf_s$.

A key advantage of the proposed setup is the intrinsic mutual coherence of the output optical waveforms with the CW seed laser. This property can be exploited to achieve direct generation of RF arbitrary waveforms on a carrier by simply heterodyning the output optical field with the seed laser. This is similar to the technique used above for characterization of the high-resolution complex-field optical spectra at the FSL output. In particular, the resulting heterodyne beating term in the photocurrent can be expressed as (see Methods):

$$I_{\text{out}}^h(t, m) \propto \left|\tilde{e}_{\text{in}}\left(\frac{f_s}{\tau_c} t\right)\right| \times \cos\left(2\pi m \frac{t}{\tau_c} + \arg\left[\tilde{e}_{\text{in}}\left(\frac{f_s}{\tau_c} t\right)\right]\right) \quad (6)$$

which corresponds to an RF waveform, whose temporal variations of amplitude and phase are set by the FT of the input modulation signal. The carrier frequency of the $m$-th waveform is equal to $m/\tau_c$ and is directly proportional to $m/f_s$, the evaluation time of the waveform.

To demonstrate this capability, we set the FSL system to work in the configuration with $f_s = 9/\tau_c = 84.91$ MHz ($\tau_c = 106$ ns). The required input modulation signal to be applied on the CW light wave through the AOM is determined by calculating the FT of the desired output temporal waveform (or the reversed one, depending on the sign of $f_s$). After heterodyning with a fraction of the seed laser, the output train of waveforms is measured with a fast photodiode and oscilloscope (25-GHz detection bandwidth). Figure 5 shows a few examples of arbitrary RF waveforms on a carrier synthesized through the proposed method, particularly, data pulse sequences arbitrarily modulated in phase (according to a quadrature phase-shift keying format, QPSK), as well as in amplitude and phase (according to a quadratic amplitude modulation format, QAM). Each of the synthesized data sequences is ~2 ns long.

## Discussion

We have demonstrated a new concept for arbitrary broadband complex-field optical spectral shaping, based on time-to-frequency mapping in a frequency shifting loop (FSL). The

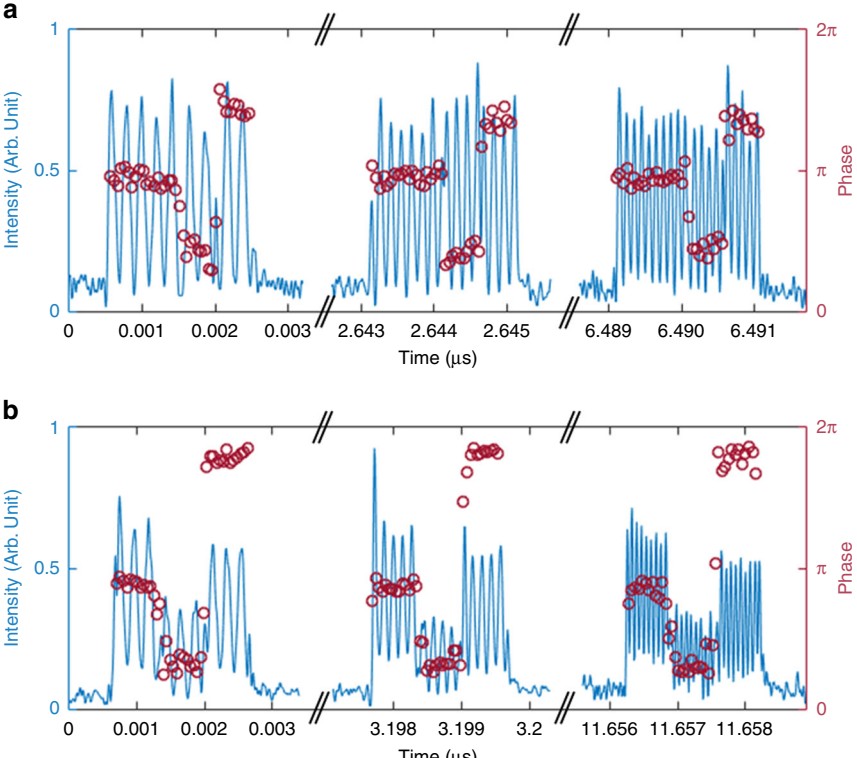

**Fig. 5** Demonstration of RF AWG on a carrier. **a** Examples of arbitrary phase modulated signals on a carrier (QPSK, 8 Gbits/s). The carrier frequencies are 5 GHz, 7 GHz, and 10 GHz from left to right, respectively. **b** examples of arbitrary amplitude and phase modulated signals on a carrier (8 QAM, 10 Gbits/s). The carrier frequencies are 5 GHz, 8 GHz, and 15 GHz from left to right, respectively

system, which involves a single CW laser, a low-bandwidth RF generator, and standard fiber-optics components, is particularly simple to implement and is easily reconfigurable. Contrary to most optical spectral shaping systems, it does not require ML lasers or fast electronics. Moreover, the proposed method offers a frequency resolution that is orders-of-magnitude higher than that of conventional optical pulse-shaping techniques. This new feature of frequency shifting loops adds to a set of previously reported capabilities of these systems, including real-time integer/fractional Fourier transformation[35,37], arbitrary frequency chirp generation[36], or generation of pulse trains with arbitrary repetition rates[32,38].

Using the experimental setup demonstrated here, we report a frequency resolution that can be adjusted by the value of the frequency shift in the FSL, from ~10 MHz to ~80 MHz. We have reported two practical applications of this system: RF AWG on a carrier, which provides arbitrary signals on an RF carrier whose frequency increases waveform to waveform in the generated output train, and baseband RF AWG, which can provide repetitive identical waveforms. The bandwidth and the TBP of the generated waveforms exceed 25 GHz and 300, respectively (both values limited by the detection equipment). It is noteworthy that our system bridges the gap between electronic-based RF AWG methods, which show high frequency resolution but limited bandwidth (<10 GHz), and conventional optical pulse-shaping techniques, which provide ultra-broad bandwidth (>THz), but poor frequency resolution (>GHz). Notice that high-speed (wideband) RF waveforms with durations above the nanosecond regime, corresponding to MHz resolutions, have been synthesized through conventional spectral optical shaping combined with frequency-to-time mapping induced by chromatic dispersion[39]. These methods are, however, strictly limited to synthesizing RF waveforms, i.e., just the temporal intensity profile

of the resulting optical waveforms. The pulse-shaping scheme proposed here allows the direct synthesis of both complex-field optical signals and RF waveforms with similar specifications to those of the dispersive method, e.g., in terms of frequency resolution and bandwidth, while entirely avoiding the need for a ML laser and bulky chromatic dispersion devices.

The parameters of the demonstrated scheme could be also easily modified to match the specifications of different potential applications. For instance, the frequency resolution could be improved to nearly arbitrary values by reducing the value of $f_s$, and correspondingly increasing the travel time $\tau_c$. On the other hand, increasing the value of $f_s$ would enable the synthesis of larger bandwidths (>100 GHz), and this could be practically achieved by use of single-sideband electro-optics modulators in the loop, instead of acousto-optic frequency shifters[38]. In our present experimental setup, the TBP is close to 300 (limited by the detection bandwidth). Ultimately, by pre-compensating the slight un-flatness of the generated spectral components, the system could be designed to achieve a TBP above 1000, consistently with previous demonstrations of FSL systems aimed at other different applications[36]. However, increasing the frequency shift or increasing the number of light round-trips in the FSL, $N$ (i.e., the waveform TBP), both require expanding the bandwidth of the TBPF in the FSL cavity. In this case, the ASE emitted by the amplifier might become a limiting factor. As detailed in the Methods section, a convenient figure of merit of our OAWG concept, in regard to noise performance, is the ratio between the power of the frequency comb produced when the FSL is seeded with a CW laser, and the total ASE power over the whole TBPF bandwidth. Since this signal-to-noise ratio (SNR) scales as the inverse of the TBPF bandwidth (see Methods), increasing $N$ or $f_s$ in the FSL system would inherently degrade the SNR of the synthesized waveforms. This effect could,

however, be mitigated by decreasing the passive losses of the FSL, and/or through the use of noise-optimized amplifiers.

In conclusion, owing to its flexibility and unique set of features, the optical waveform shaping technique introduced herein should fulfil the stringent requirements for a wide range of applications in fundamental physics, telecommunications, microwave photonics, etc.

## Methods

**Time-to-frequency mapping in injected FSLs**. Here we provide a mathematical proof of the mapping of the complex temporal signal at the input of the loop (duration: $\tau$), to the optical field spectrum measured at its output. For analysis of the output signal, we define $w_m(t)$ as a temporal window function centered at time $m/f_s$ ($m = 1, 2, 3, \ldots$) and with a duration $\sim 1/f_s$. In other words, the parameter $m$ defines the evaluation time of the outcoming signal, in steps of $1/f_s$.

We define $e_{in}(t)$ as the complex RF modulation signal imprinted by the AOM onto the monochromatic laser, so that the electric field at the input of the loop can be written as: $E_{in}(t) = e_{in}(t)e^{i2\pi f_0 t}$ (analytical representation, with $e_{in}(t)$ being the temporal complex envelope). As derived in the main text (Eqs. (1) and (2)), when the product $f_s\tau_c$ is equal to an integer ($p$), the electric field at the output of the FSL is given by the following expression:

$$E_{out}(t) = \gamma \sum_{n=0}^{N} E_{in}(t - n\tau_c)e^{i2\pi n f_s t}. \tag{7}$$

The number of replicas of the input is set by the spectral bandwidth of the TBPF, $Nf_s$. The slowly varying envelope of the output field $e_{out}(t)$, defined by $E_{out}(t) = e_{out}(t)e^{i2\pi f_0 t}$, is given by:

$$e_{out}(t) = \gamma \sum_{n=0}^{N} e_{in}(t - n\tau_c)e^{i2\pi n(f_s t - f_0\tau_c)}. \tag{8}$$

The envelope of the output field along the temporal window $w_m$ is simply: $e_{out}(t, m) = w_m(t) \times e_{out}(t)$. Consequently:

$$e_{out}(t, m) = \gamma w_m(t) \times \sum_{n=0}^{N} e_{in}(t - n\tau_c)e^{i2\pi n(f_s t - f_0\tau_c)}. \tag{9}$$

We have assumed that the temporal variations of $e_{in}$ are slower than $\tau_c$, which means that $e_{in}(t)$ can be considered as constant over the duration of the window function $w_m$ (i.e., over $1/f_s = \tau_c/p$). Therefore:

$$e_{out}(t, m) = \gamma w_m(t) \times \sum_{n=0}^{N} e_{in}(m/f_s - n\tau_c)e^{i2\pi n(f_s t - f_0\tau_c)}. \tag{10}$$

For simplicity reasons, the exponential term $f_0\tau_c$ in Eq. (10) can be ignored, as this simply represents a change in the origin of time. The baseband spectrum measured at time $m/f_s$ is: $\tilde{e}_{out}(f, m) = FT(e_{out}(t, m))$, where FT is the Fourier transform operator, defined as: $FT(s(t)) = \tilde{s}(f) = \int_{-\infty}^{+\infty} s(t)e^{-i2\pi ft} dt$. Owing to the convolution theorem of the FT:

$$\tilde{e}_{out}(f, m) = \gamma \tilde{w}_m(f) \otimes \sum_{n=0}^{N} e_{in}(m/f_s - n\tau_c)\delta(f - nf_s)$$
$$= \gamma \tilde{w}_m(f) \otimes \left[ e_{in}\left(\frac{m - f\tau_c}{f_s}\right) \sum_{n=0}^{N} \delta(f - nf_s) \right]. \tag{11}$$

Once more, since the frequency bandwidth of $e_{in}(t)$ is smaller than $1/\tau_c$, $e_{in}\left(\frac{m - f\tau_c}{f_s}\right)$ shows negligible variations when $f$ varies by $f_s$, i.e., along the width of $\tilde{w}_m(f)$. Then:

$$\tilde{e}_{out}(f, m) = \gamma e_{in}\left(\frac{m - f\tau_c}{f_s}\right) \times \left[ \tilde{w}_m(f) \otimes \sum_{n=0}^{N} \delta(f - nf_s) \right]. \tag{12}$$

The term in brackets depends only on the choice of the window function. For convenience of analysis, by properly designing the specific window shape, this term can be approximated by a flat-top function extending along the frequency range $0 < f < Nf_s$, i.e., this is a constant term that vanishes when $f < 0$ or $f > Nf_s$. Therefore, the instantaneous output spectrum at time $m/f_s$ can be expressed as follows:

- for $f < 0$ or $f > Nf_s$

$$\tilde{e}_{out}(f, m) = 0 \tag{13}$$

- for $0 < f < Nf_s$

$$\tilde{e}_{out}(f, m) \propto e_{in}\left(\frac{m - f\tau_c}{f_s}\right) \tag{14}$$

This latest equation corresponds to the mapping of the input temporal complex-modulation signal into the optical field spectrum of the output signal, as illustrated in Fig. 1. The time-to-frequency scaling coefficient is equal to $f_s/\tau_c$.

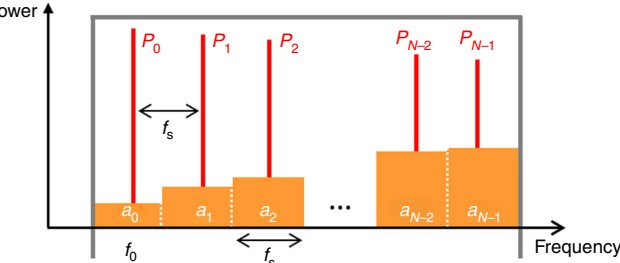

**Fig. 6** Optical spectrum in the FSL seeded by a CW laser. The FSL is seeded by a CW monochromatic laser at frequency $f_0$. A frequency comb is created in the loop (FSR: $f_s$), in red. The ASE background is shown in orange. The filter transmission function is assumed to exhibit a flat-top profile (in gray)

**Electric field at the output of the FSL**. Suppose now that the whole input signal has entered the FSL, i.e., $m > f_s\tau$. In this case, the output electric field envelope within the temporal window $w_m$ is given by the inverse FT of Eq. (14):

$$e_{out}(t, m) \propto \tilde{e}_{in}\left(\frac{f_s}{\tau_c}t\right)e^{i2\pi m \frac{t}{\tau_c}}. \tag{15}$$

This expression shows that the envelope of the output signal is simply equal to the spectrum of the input envelope mapped into the temporal domain, and multiplied by a linear phase term. The output electric field describing the waveform emitted at $m/f_s$ can then be written as

$$E_{out}(t, m) \propto \tilde{e}_{in}\left(\frac{f_s}{\tau_c}t\right)e^{i2\pi\left(f_0 + \frac{m}{\tau_c}\right)t} \tag{16}$$

and the corresponding photocurrent (at the output of the photodetector) is:

$$I_{out}(t, m) \propto \left|\tilde{e}_{in}\left(\frac{f_s}{\tau_c}t\right)\right|^2. \tag{17}$$

Suppose now that the train of waveforms, generated in the FSL, is recombined with a fraction $\beta$ of the seed laser (see Fig. 1), and sent to the photodetector. The photocurrent during the temporal window $w_m$ can be expressed as follows:

$$I_{out}^h(t, m) \propto \left|\beta e_0 e^{i2\pi f_0 t} + \tilde{e}_{in}\left(\frac{f_s}{\tau_c}t\right)e^{i2\pi\left(f_0 + \frac{m}{\tau_c}\right)t}\right|^2 \tag{18}$$

where the superscript "h" stands for "heterodyne". Assuming that the intensity of the seed laser is much larger than that of the output waveform, the photocurrent expression can be rewritten as

$$I_{out}^h(t, m) \propto (\beta e_0)^2 + \beta e_0 \left|\tilde{e}_{in}\left(\frac{f_s}{\tau_c}t\right)\right| \times \cos\left(2\pi m\frac{t}{\tau_c} + \arg\left[\tilde{e}_{in}\left(\frac{f_s}{\tau_c}t\right)\right]\right). \tag{19}$$

**ASE and signal-to-noise ratio**. In the following, we provide a simple model of an FSL in the steady-state, in order to provide an estimation of the SNR of the proposed OAWG process. The main source of noise in the system is the amplified spontaneous emission (ASE) emitted by the amplifier placed inside the FSL, because similarly to the seed signal, the ASE is repeatedly frequency-shifted and amplified in the loop. To get a better understanding of the effects of ASE without excessive complexity, we address the steady-state case, where the FSL is seeded with a CW laser (frequency: $f_0$, power: $P_0$). The light in the FSL is a comb of $N$ optical frequencies, starting at $f_0$, and spaced by $f_s$ (Fig. 6). We assume that the spectral transfer function of the TBPF in the FSL exhibits a flat-top profile with a total bandwidth of $Nf_s$. We define $P_n$, the power of the comb line at frequency $f_0 + nf_s$, and $a_n$ the power of the ASE integrated in a frequency band of width $f_s$, centered around $f_0 + nf_s$.

We also define $G$ as the power amplification coefficient of the amplifier, and $T$ as the single-pass transmission coefficient of the FSL, excluding the amplifier. $P_n$ and $a_n$ satisfy the following recurrence relationships:

$$\begin{aligned} P_n &= GT \times P_{n-1} \\ a_n &= GT \times a_{n-1} + a_0, \end{aligned} \tag{20}$$

where $a_0 = n_{sp}hf_0(G - 1)f_s$ is the ASE power generated by the amplifier in a single polarization state (here, we are using polarization-maintaining fibers), and a frequency bandwidth of $f_s$. In this latest expression, $h$ is the Planck constant, and $n_{sp}$ is the spontaneous emission factor of the amplifier ($n_{sp} > 1$)[40]. As per the design conditions of the proposed OAWG concept, we consider the case in which the amplifier just compensates for the losses in the FSL cavity, i.e. when $GT \approx 1$. A convenient estimation of the SNR of the OAWG process can be given by the ratio between the total comb power (signal), divided by the total ASE power in the FSL (noise). The total comb power ($P_{comb}$) and the ASE power ($P_{ASE}$) are, respectively,

equal to:

$$P_{comb} = \sum_{n=0}^{N-1} P_n = NP_0$$

$$P_{ASE} = \sum_{n=0}^{N-1} a_n = \frac{N(N+1)}{2} a_0,$$

(21)

and the signal-to-noise ratio is then: $SNR \approx 2P_0/(Na_0)$. Interestingly, the SNR scales as the inverse of the spectral bandwidth $Nf_s$. In practice, typical values of $P_0$ range from a few μW to a few tens of μW. Stronger injection results in an exponentially decreasing comb envelope, due to gain saturation of the amplifier[41]. For $P_0 = 1$ μW, $f_s = 80$ MHz, $n_{sp} = 2$, $G = 1/T = 10$, and $N = 300$, the SNR is about 35. Notice that the SNR could be substantially improved by increasing the seed power (i.e., by increasing the saturation power of the amplifier), by reducing the losses of the loop (i.e., increasing $T$ so as to reduce $G$), and/or by using a low noise optical amplifier.

**Architecture of the FSL.** The frequency shifting loop (FSL) comprises an erbium-doped fiber amplifier (EDFA, Pritel, PMFA-15), a tunable bandpass filter (TBPF, Yenista, XTM-50), and one or two fiber acousto-optic frequency shifters (AOFS, AA Opto-electronic, MT80-IIR30-Fio) (Fig. 1). The role of the EDFA is to compensate for the passive losses of the FSL. In practice, the EDFA gain is about ~10 dB. In a first configuration, two AOFS are inserted in the loop: the first one generates a positive frequency shift (+80 MHz ± 5 MHz)), and the second one, a negative frequency shift (−80 MHz ± 5 MHz). The frequencies driving the two AOFS are set in such a way that the net frequency shift per round-trip is $f_s = 1/\tau_c$ = 9.482 MHz (The FSL round-trip propagation time is: $\tau_c = 105$ ns). The loop also contains an optical isolator to prevent back-reflection of the light. All fiber-optics components are polarization-maintaining. The role of the TBPF is two-fold: first to control the spectral bandwidth of the light field in the loop (i.e., the maximum number of round-trips $N$), and second, to limit the influence of the ASE from the EDFA. A narrow-linewidth (<0.1 kHz) CW laser (OEwaves, HI-Q™ 1.5 Micron Laser), delivering 10 mW of optical power at 1550.0 nm is split by means of a 3 dB coupler. The first port is sent to an acousto-optics modulator (AA Opto-electronic, MT80-IIR30-Fio, the AOM, in Fig. 1), driven by an arbitrary function generator (Keysight, 33600A). The modulation signal $e_{in}(t)$ is defined according to the desired signal at the output of the FSL, and multiplied by a single-frequency tone with a frequency $f_m = 80$ MHz. Recall that the AOM induces a frequency shift of $f_m$ on the CW laser. A variable optical attenuator (not shown in Fig. 1) is used to control the power injected in the FSL. The second port is used as a reference arm (local oscillator) and can be recombined with the FSL output (self-heterodyning). The light circulating in the loop is extracted by means on a 2% Y-coupler. The intensity is detected by a fast photodiode (20 ps rise-time), and recorded by a 3.5 GHz, or by a 25 GHz-bandwidth real-time digital oscilloscope. All measured waveforms reported in this article are single-shot time traces (no averaging is performed).

This first configuration is particularly well suited to measure the amplitude and phase of the optical spectrum of the output waveforms, since heterodyning with the CW seed laser induces a down-conversion from the optical to the RF domain. The first configuration is also used for generating long (~100 ns) arbitrary waveforms. In a second configuration, the same setup is used, except that a single AOFS is inserted in the FSL. It is driven at $f_s = 9/\tau_c = 84.91$ MHz ($\tau_c = 106$ ns). This configuration is used to demonstrate high bandwidth AWG.

## Data availability

The data that support the findings of this study are available on request from the corresponding author H.G.C.

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

## Acknowledgements

This research was supported by the Agence Nationale de la Recherche (grant number ANR-14-CE32-0022). We acknowledge Tektronix France for the loan of a 25 GHz-bandwidth real-time digital oscilloscope.

## Author contributions

J.A. and H.G.C. initiated the project, C.S. carried out the experiments, C.S. and H.G.C. carried out the theoretical analysis, with feedback from J.A. All authors contributed to the manuscript writing.

## Competing interests

The authors declare no competing interests.
