## [Peer Review File · Nature Communications]

Reviewers' comments:

Reviewer #1 (Remarks to the Author):

This manuscript addresses a simple technique, easy to implement, which has a high potential interest for generating arbitrary coherent optical waveforms. This can have a high impact on the development of lightwave technologies. I was positively seduced by the proposition and was very excited to know more about it when I started to read the manuscript.

I am deeply sorry to say to the authors that I was largely disappointed by the description of the technique and their report. The concept can be hardly understood and I have to admit that even after 2 thorough readings I could not grasp it entirely. It is not acceptable that a publication in a flagship journal turns into a challenge to guess what the authors want to mean exactly. This paper has the form of a standard publication, like a simple scientific report. The storytelling must be much better developed.

The authors must really make the intellectual effort to figure out how the concept can be grasped by a reader who has absolutely no experience and no intuition about the method. Explaining with only pure mathematical proofs is not sufficient for such an ambitious publication and "hand waving" style clarification must be initially brought to guide the reader's intuition. For instance, it is absolutely not obvious why the spectrum of the output signal is a scaled copy of the temporal waveform of the input signal. After 2 readings it is not intuitively clear for me. With a little effort, the authors could find a simple signal waveform (even unrealistic, purely academic) that would explain the reasons.

On many aspects the authors could not make the effort to take the place of the reader. There are concepts that remain totally opaque to an external specialist (e.g. impossible to understand why there are 2 time variables in the expressions (t, t')). There are a large number of variables and their definition is not repeated after 2-3 pages (and in the methods section), which makes the reading very tedious. Acronyms only defined for this publication are used in figure captions and this has to be avoided for Nature manuscripts.

I must also point out a couple of major issues in the manuscript: the conditions are not clearly set at the beginning of the explanations. For instance this is only in the methods section that the reader is informed of some limits on the waveform duration. In addition the main text must be self-contained and a prior reading of the methods section should not be made to understand its content. I don't see the point to mention all over the main section the implementation of 2 different configurations which are at last eventually explained in the methods section: this is really a minor change and this must not be addressed like 2 configurations, since it just brings confusion.

The authors also skipped nearly entirely the limitations in their description and even more in their conclusions: this is unacceptable for such an ambitious publication. For instance the authors keep entirely silent about the impact of noise. Considering their presented measurements we can easily observe that it is fairly important. I can understand that the presence of frequency shifting makes a closed-loop gain higher than unity possible, without self-oscillation, but the EDFA noise will sum up incoherently at each roundtrip. Probably it will cumulate more at one side of the spectrum, due to the presence of the filter (the noise is spectrally pushed on one side by the frequency shift). Conditions on the signal waveform (e.g. duration) are never clearly set.

Finally it remains unclear what kind of signal is obtained at the output: is it a repetitive waveform (periodic) of the transformed input signal? Does it need to be gated if a pure transformed waveform has to be extracted? If not, what is the purpose of the windowing function in the theoretical description?

My collegial recommendation to the authors is that they have certainly a very valuable concept to deliver to the community, but they must make the intellectual exercise to rebuild their explanation to make it easily seized by a totally uninformed reader. Simple solutions are often the best and the community would love to know it. In the present form I am afraid that this brilliant idea won't attract the attention it deserves.

Reviewer #2 (Remarks to the Author):

Title: Programmable broadband optical field spectral shaping with MHz resolution using a simple frequency shifting loop

Comments:

The authors reported a programmable broadband optical field spectral shaping method based on a frequency shifting loop. The frequency shifting loop is used to generate a frequency comb whose shape is manipulated according to the time shape of the intensity modulated signal. The method is interesting since the spectral resolution can reach MHz level. However, before accepting the paper for publication, there are still some issues which must be figured out.

1. The system proposed in this paper uses the Fourier transformation reported in [37] to do time-frequency mapping, and use a similar structure in [36] to improve the spectral manipulation resolution. It seems the only novelty of the system is that the system can generate a frequency comb with arbitrary shape. So, please confirm that what the main differences between the proposed method and Ref. [36] and [37] are, and what the main novelty of the proposed system is.
2. The time-bandwidth product (TBP) based on this method is around 100, which is too low for practical applications, making the method unattractive. Is there any possible way to increase the TBP, and what is the maximum value that the method can achieve?
3. Actually, the relationship of the f_s and the τ_c is very rigorous, but two AOMs, an EDFA, a tunable optical filter are employed in the frequency shifting loop. So how do you realize this system, which consists of a polarization maintaining optical filter, a polarization maintaining AOFS, and a polarization maintaining EDFA? What is the operating status of the EDFA? Can you provide your actual experimental devices? In addition, what is the effect of the time jitter of the loop on the experimental results?
4. As shown in Fig. 2c, why are the bandwidths of the selected output waveforms different? The bandwidth of the first waveform seems to be smaller than that of the other two. What is the reason and is there a relationship between the bandwidth of the selected output waveform and the time?
5. In Fig. 3, the authors use scattered points to represent the generated waveform, why not use curves with a large number of points? Please provide the root mean square error between the generated and the designed waveforms.
6. In Fig. 4, a unique modulated temporal signal corresponds to the desired RF signal. Thus for a better description, I suggest adding the modulated temporal signal as the inset in Fig. 4.

Reviewer #3 (Remarks to the Author):

In this publication by Schnebelin et al, a system is demonstrated based on an optical recirculating loop which is claimed to generate programmable arbitrary optical fields. The usefulness of the proposed system can be debated, however, owing to various issues I point out below, this paper is not suitable for publication in the journal. My main points are –

1. The authors seem to arbitrarily use optical shaping, OAWG and what they do interchangeably. In reality, it is extremely evident that this work is only shaping low frequency components of the optical envelope and then photo-detecting it. Optical arbitrary waveform generation is completely incorrect. This is RF waveform generation through optical means.
2. Effectively, what the authors achieve is time-stretch of a waveform that is originally imparted on the optical carrier. All information content that the final waveform has is effectively imparted in the initial modulation of the CW carrier. So fundamentally, the best achievable TBP is again limited by the input TBP of the modulating waveform. This system is not overcoming any fundamental limitation in that regard. However, what is done is, since the time-stretch is in the Fourier domain,

there seems to be some scaling of time. This however is limited in addition by the circulating time in the loop and related systemic issues.

3. A substantial amount of this paper is poorly written with confusing details. On one hand they claim 100ns waveform (corresponding to 10MHz resolution) with 25GHz detection bandwidth, this should correspond to a TBP of 2500, but then in the paper they claim a 300 TBP.

4. My most vehement opposition to the paper is that. There is absolutely no result in this paper which cannot just be synthesized by a commercially available RF Arbitrary waveform generator. I will not talk of synthesis of optical arbitrary waveforms since this work is not even remotely connected to that. A minimal figure of merit I expect from RF photonic techniques is that there should at least be one result which is not possible with existing standard RF equipment. This work completely fails in that regard.

I wish I had more positive things to say, but this paper is completely unsuitable for publication in nature communications.

Reviewer #1:

This manuscript addresses a simple technique, easy to implement, which has a high potential interest for generating arbitrary coherent optical waveforms. This can have a high impact on the development of lightwave technologies. I was positively seduced by the proposition and was very excited to know more about it when I started to read the manuscript.

I am deeply sorry to say to the authors that I was largely disappointed by the description of the technique and their report. The concept can be hardly understood and I have to admit that even after 2 thorough readings I could not grasp it entirely. It is not acceptable that a publication in a flagship journal turns into a challenge to guess what the authors want to mean exactly. This paper has the form of a standard publication, like a simple scientific report. The storytelling must be much better developed.

The authors must really make the intellectual effort to figure out how the concept can be grasped by a reader who has absolutely no experience and no intuition about the method. Explaining with only pure mathematical proofs is not sufficient for such an ambitious publication and "hand waving" style clarification must be initially brought to guide the reader's intuition. For instance, it is absolutely not obvious why the spectrum of the output signal is a scaled copy of the temporal waveform of the input signal. After 2 readings it is not intuitively clear for me. With a little effort, the authors could find a simple signal waveform (even unrealistic, purely academic) that would explain the reasons.

On many aspects the authors could not make the effort to take the place of the reader.

We thank the referee for this very important remark. We completely agree that our point should be made clear. Therefore we have completely rephrased the theoretical description section. We have tried to make it simpler, and to explain things in a natural and intuitive way, while keeping the important mathematical formulas. Especially we have replaced the original Fig.1 by a new one. We believe that this new figure renders our description much more intuitive. In our opinion, it enables to understand in a simple manner the fundamental property of our technique, namely the time-to-frequency mapping property. Moreover this new figure also highlights the intrinsic design trade-offs of the technique (duration and bandwidth of the input signal, role of the tunable bandpass filter...). To help the reader in understanding the basis of our proposed method, the descriptions of the principle of operation in the main text of the manuscript are based on the intuitive illustrations in Fig. 1, with some additional, though minimal, mathematical support. A full mathematical analysis of the proposed concept is subsequently provided in the Methods section.

There are concepts that remain totally opaque to an external specialist (e.g. impossible to understand why there are 2 time variables in the expressions (t, t')). There are a large number of variables and their definition is not repeated after 2-3 pages (and in the methods section), which makes the reading very tedious. Acronyms only defined for this publication are used in figure captions and this has to be avoided for Nature manuscripts.

Regarding the pertinence of defining two different time scales, we agree with the referee that the mathematical expressions could be simplified, enabling a more straightforward interpretation. We have now removed the variable t' , and we now refer to the successive waveforms at the

output of the frequency shifting loop by an integer number (m). Regarding the acronyms specific to this study (such as FSL), they have all been removed from Fig. 1.

I must also point out a couple of major issues in the manuscript: the conditions are not clearly set at the beginning of the explanations. For instance this is only in the methods section that the reader is informed of some limits on the waveform duration. In addition the main text must be self-contained and a prior reading of the methods section should not be made to understand its content.

Again, we thank the referee for this important point. In response, we have made the needed changes to keep the manuscript as readable as possible, without incorporating too many mathematical developments into the body text. We believe that the new Fig. 1 enables to understand, in a straightforward manner, the requirements and the design trade-offs and limitations of the proposed system, e.g., in terms of input signal duration and bandwidth. Basically, the whole temporal input signal is mapped along the output optical spectrum, as long as the tail of the input signal has not been filtered out by the tunable bandpass filter, and the tail of the signal has already entered the FSL. Therefore, the discussions on the design relationships and limitations mentioned by the referee are now incorporated into the main body of the article, while being supported by the new Fig. 1. Detailed mathematical developments have been left in the Methods section.

I don't see the point to mention all over the main section the implementation of 2 different configurations which are at last eventually explained in the methods section: this is really a minor change and this must not be addressed like 2 configurations, since it just brings confusion.

According to the referee's suggestion, we have removed this distinction between the two previous configurations.

The authors also skipped nearly entirely the limitations in their description and even more in their conclusions: this is unacceptable for such an ambitious publication. For instance the authors keep entirely silent about the impact of noise. Considering their presented measurements we can easily observe that it is fairly important. I can understand that the presence of frequency shifting makes a closed-loop gain higher than unity possible, without self-oscillation, but the EDFA noise will sum up incoherently at each roundtrip. Probably it will cumulate more at one side of the spectrum, due to the presence of the filter (the noise is spectrally pushed on one side by the frequency shift).

We agree with the referee that this question is very important. Indeed, as pointed out by the referee, the amplifier brings extra noise (amplified spontaneous emission, or ASE) at each roundtrip in the frequency shifting loop. The effect in the spectral (resp. temporal) domain is to degrade the fidelity of the generated output spectrum (resp. waveform). To evaluate the influence of this extra noise, we have developed a simple and intuitive model in the steady state, when the FSL is seeded by a monochromatic CW (i.e., un-modulated) laser. In this case, the output spectrum consists of a comb of optical frequencies separated by f_s , and of an ASE background. Then, it is relatively straightforward to calculate the power of the comb lines, the spectral power density of the ASE, and to define a signal-to-noise ratio (SNR) as the total comb power divided by the total ASE power. We have shown that for the typical experimental conditions we consider, this SNR is above 35, and we propose some solutions to further increase this value. This model has been added to the Methods section, including a new supporting figure

(Fig. 6). (Notice that a more complete model of the spectral features at the output of the FSL has been submitted by the LIPhy team and should be published shortly.) Finally, regarding the experimental results, a good estimation of the fidelity of the generated spectra is given by the RMS error signal between the input temporal modulation signal, and the generated spectra (see Referee #2). These values have been calculated, and added to the captions of Fig.3.

Conditions on the signal waveform (e.g. duration) are never clearly set.

Again, we thank the referee for bringing up this important point. We believe that the new Fig. 1 clearly shows in an intuitive manner, the intrinsic limitations of the OAWG technique in terms of input signal duration. Additional sentences have also been added to the text to emphasize these limitations.

Finally it remains unclear what kind of signal is obtained at the output: is it a repetitive waveform (periodic) of the transformed input signal? Does it need to be gated if a pure transformed waveform has to be extracted? If not, what is the purpose of the windowing function in the theoretical description?

We apologize for this lack of clarity. Indeed the output consists of a repetitive train of optical waveforms (now labelled by the integer m) with identical intensity, and linearly increasing frequency. This is true as long as the tail of the input signal has already entered the FSL, and before the head of the input signal has been filtered out by the band-pass filter. This limitation is now rendered clear in Fig. 1 and apparent in Eq. 3 and Fig. 4. Notice that we have redrawn Fig.4 to show on the same plot (Fig. 4 a, top and middle), the input modulation signal (phase and amplitude), and the output train of waveforms. In order to avoid any further ambiguity in this regards, we added in the text a few sentences to explain this property. See for instance, page 7: "Briefly, from Eq. (2), it can be shown that the resulting time-domain signal at the output of the FSL consists of a train of consecutive waveforms separated by $1/f_s$, here labelled by m ($= 1, 2, 3 \dots$).", or: "For analysis of the output signal, we define $w_m(t)$ as a temporal window function centered at time m/f_s ($m = 1, 2, 3, \dots$) and with a duration $\sim 1/f_s$. In other words, the parameter m defines the evaluation time of the outcoming signal, in steps of $1/f_s$." in the Methods section.

Reviewer #2:

1. The system proposed in this paper uses the Fourier transformation reported in [37] to do time-frequency mapping, and use a similar structure in [36] to improve the spectral manipulation resolution. It seems the only novelty of the system is that the system can generate a frequency comb with arbitrary shape. So, please confirm that what the main differences between the proposed method and Ref. [36] and [37] are, and what the main novelty of the proposed system is.

The system described in this manuscript is similar to the one reported in [37]; however, we report new findings on the fundamental behavior of this laser system which allows us to demonstrate a new, important capability of the system, not previously anticipated, namely the capability to realize a mapping of the input complex (amplitude and phase) temporal waveform

into the output frequency spectrum. This leads to entirely different applications for the system. To be more specific, in [37], we have shown that the FSL set in a condition where the product $f_s \tau_c$ is an integer number, realizes a mapping from the input spectrum to the output time trace. We have demonstrated this property for input signals with infinite duration, and constant instantaneous spectrum. In this manuscript, we have extended this concept into three new, different directions: (i) we have demonstrated, for the first time, that the system enables the mapping of the temporal input signal to the instantaneous frequency spectrum of the output signal; (ii) using this finding, the system is now used for signal generation purposes, rather than for signal analysis or processing, as in [37]; and (iii) it is not restricted to infinite input signals, . Notice that the key finding reported here, as defined in (i), cannot be directly inferred from the analysis and results in paper [37]. In this previous paper, we demonstrated that the FSL system can be set to implement a mapping of the amplitude (or intensity) frequency spectrum of the input signal along the time domain, i.e., frequency-to-time mapping, with no discussions on whether this mapping involves the spectral phase profile as well (or how this is affected by the process). The results and conclusions in our present paper show that the frequency-to-time mapping that was demonstrated for the amplitude spectrum in Ref. [37] also applies to the signal spectral phase profile, such that the system inherently implements a mapping of the input temporal wave profile along the frequency domain as well, i.e., a full exchange of the time and frequency-domain profiles, both in amplitude and in phase, from the input to the output of the FSL system. This general property is predicted, analyzed and demonstrated in our new submission for the very first time. We believe that the new version of the manuscript makes this central novelty more evident. On the other hand, the work in [36] is focused on a completely different situation, where the system is set in the vicinity of the integer condition ($f_s \tau_c = \text{integer} + \varepsilon$). Then, when seeded by a CW monochromatic laser, the FSLs provides a comb of optical frequencies with a built-in quadratic spectral phase. The corresponding time trace is a train of optical chirped waveforms.

2. The time-bandwidth product (TBP) based on this method is around 100, which is too low for practical applications, making the method unattractive. Is there any possible way to increase the TBP, and what is the maximum value that the method can achieve?

In the experiments described in the manuscript, we have demonstrated a TBP as large as 300 (see for instance Fig. 3, c, or Fig. 4). For instance in Fig. 3, c, the spectral bandwidth shown on the plot is 2 GHz for a value of f_s close to 9.5 MHz, which corresponds to a TBP close to 220. In Fig. 4, we demonstrate the generation of arbitrary waveforms with 3.5 GHz bandwidth, and 100 ns duration, corresponding to a TBP of 350. More importantly, we have added a subplot to Fig. 3, (a), that proves the possibility of generating (and controlling) about 1000 frequency components. Therefore, theoretically, the maximum TBP that we can achieve using the specific setup demonstrated in our paper, is as high as 1000. However, experimentally, the flatness of the generated spectrum is not perfect, and this should be properly compensated for OAWG with such a large number of frequency components. Here, we restricted ourselves to a relatively smaller TBP (i.e., 300), where the spectral un-flatness doesn't need to be compensated.

Still, we do not entirely agree that this level of performance is unattractive for practical applications. This value is similar to the performance that can be achieved from commercial optical pulse shapers, which provide a maximum TBP close to 500 (frequency resolution in the 10 GHz range, with a total input bandwidth of 5 THz). But again, the frequency resolution of the

technique described in our manuscript is orders of magnitude smaller (80 MHz, down to 10 MHz) than the frequency resolution of present commercial optical pulse shapers.

3. Actually, the relationship of the f_s and the τ_c is very rigorous, but two AOMs, an EDFA, a tunable optical filter are employed in the frequency shifting loop. So how do you realize this system, which consists of a polarization maintaining optical filter, a polarization maintaining AOFS, and a polarization maintaining EDFA? What is the operating status of the EDFA? Can you provide your actual experimental devices? In addition, what is the effect of the time jitter of the loop on the experimental results?

We thank the reviewer for this interesting point. Following his suggestion, in the Methods section, we provide the manufacturer and the model of the different optical components involved in our set-up (laser, amplifier, AOFS, filter...). We also indicate in the Methods that the role of the EDFA is only to compensate for the passive losses of the FSL, and provides a moderate gain (typically ~ 10 dB). Finally, regarding the timing jitter, the effect of the fluctuations of the length of the loop (such as vibrations or thermal drifts), is to induce an overall temporal delay on the output time trace (See for instance Guillet de Chatellus, H., et al. Theory of Talbot lasers, Phys. Rev A 88, 033828 (2013).). However, the time scale of such fluctuations is in the ms, i.e. much longer than the duration of the output sequence, typically. the μs range (see Fig 4.a). Therefore the fluctuations of the length of the loop, have no significant impact on the timing jitter of the output time trace.

4. As shown in Fig. 2c, why are the bandwidths of the selected output waveforms different? The bandwidth of the first waveform seems to be smaller than that of the other two. What is the reason and is there a relationship between the bandwidth of the selected output waveform and the time?

This is a good point, indeed. The bandwidth of the first extracted waveform is smaller, because at this time, the tail of the input modulation signal has not entered the FSL. Therefore the number of spectral components contained in the first spectrum is smaller than in the following spectra, where the whole temporal input has been loaded in the loop. We believe that the reviewer's comment was due to the fact that the explanation of the concept was not clear enough. As said, we believe that Fig. 1 now enables a better understanding of the basic principle and limitations of the technique, including the evolution of the resulting waveforms from the loop along the time domain, as described. To make things even more clear, we have also included additional explanation sentences at the end of the caption of Fig. 2.

5. In Fig. 3, the authors use scattered points to represent the generated waveform, why not use curves with a large number of points? Please provide the root mean square error between the generated and the designed waveforms.

We prefer to use a scatter plot for the experimental plots, in order to illustrate better the improved resolution offered by the spectral shaping technique. In our plots, each data point corresponds to a frequency that can be controlled in amplitude and phase. Regarding the RMS error between the designed and the generated waveforms, the corresponding values have been calculated, and added to the caption of Fig. 3.

6. In Fig. 4, a unique modulated temporal signal corresponds to the desired RF signal. Thus for a better description, I suggest adding the modulated temporal signal as the inset in Fig. 4.

We thank the reviewer for this good suggestion. The corresponding traces have been added to Fig. 4.

Reviewer #3:

In this publication by Schnebelin et al, a system is demonstrated based on an optical recirculating loop which is claimed to generate programmable arbitrary optical fields. The usefulness of the proposed system can be debated, however, owing to various issues I point out below, this paper is not suitable for publication in the journal. My main points are –

1. The authors seem to arbitrarily use optical shaping, OAWG and what they do interchangeably. In reality, it is extremely evident that this work is only shaping low frequency components of the optical envelope and then photo-detecting it. Optical arbitrary waveform generation is completely incorrect. This is RF waveform generation through optical means.

We thank the reviewer for this comment, but we disagree with it. Indeed our technique enables to control in amplitude and phase more than 300 spectral components of a light field (theoretically, up to 1000 with the demonstrated setup, cf Fig. 3, a). By properly setting the amplitude and phase of these spectral components, we generate optical arbitrary waveforms, with user-defined amplitude and phase temporal profiles. The total bandwidth of the generated waveform can exceed tens of GHz, so that our technique bridges the gap between broadband OAWG by spectral shaping techniques, and OAWG by direct complex (I/Q) temporal modulation of CW light. The generated optical spectra can be measured by heterodyne mixing with the CW laser. As a relevant application, the technique is used for RF AWG by directly photodetecting the generated optical arbitrary waveform. Nonetheless, along the paper, we have tried to make it clear that our technique is primarily an OAWG technique.

2. Effectively, what the authors achieve is time-stretch of a waveform that is originally imparted on the optical carrier. All information content that the final waveform has is effectively imparted in the initial modulation of the CW carrier. So fundamentally, the best achievable TBP is again limited by the input TBP of the modulating waveform. This system is not overcoming any fundamental limitation in that regard. However, what is done is, since the time-stretch is in the fourier domain, there seems to be some scaling of time. This however is limited in addition by the circulating time in the loop and related systemic issues.

We agree with the referee that all the information contained in the output waveform is contained in the input one. More precisely, the TBP is maintained from the input to the output. However, the unique and most important feature of the proposed concept is that the system implements a very large temporal compression process (i.e., an effective frequency bandwidth increase) from the input to the output. More precisely, on the one hand, the bandwidth of the input signal is smaller than the inverse of the round-trip time in the cavity, namely $1/\tau_c$, which corresponds to 10 MHz in our experiments. On the other hand, the bandwidth of the output

signal can be as high as N times the frequency shifting imposed in each round trip, namely, Nf_s , that is to say 25 GHz in our experiments (limited here by the detection bandwidth). This temporal compression factor enables to generate broadband output signals by using a RF waveform generator with a modest input bandwidth. To highlight further this key feature, a sentence at the end of paragraph 2.1 emphasizes the extremely large bandwidth ratio between the input and the output signals: "Notice also that the bandwidth of the output optical waveform ($\sim \tau f_s / \tau_c$) is about τf_s times, *i.e.*, orders of magnitude larger than the bandwidth of the input modulation signal ($< 1 / \tau_c$). This enables generation of broadband optical waveforms (e.g., with a bandwidth up to hundreds of GHz [36]) at the FSL output from low-bandwidth electronic input signals (typ. < 10 MHz)."

3. A substantial amount of this paper is poorly written with confusing details. On one hand they claim 100ns waveform (corresponding to 10MHz resolution) with 25GHz detection bandwidth, this should correspond to a TBP of 2500, but then in the paper they claim a 300 TBP.

We believe that we have now improved the quality and readability of the manuscript. Our experiments have been carried out under different experimental conditions, depending on the equipment available in the lab. This is why we have always provided the value of f_s and the detection bandwidth. The referee is right when he claims that combining a 10 MHz resolution (*i.e.* the value of f_s) to a fast detection chain (25 GHz bandwidth) would give a maximum TBP different to the one actually demonstrated in our experiments. In practice, it turns out that it is experimentally difficult to generate more than $N = 1000$ spectral components with a constant magnitude (see Fig. 3, a). Any attempt to expand the spectral bandwidth at the output of the FSL results in instability and ASE noise issues. In response to the referee's comment, an insight of this limitation has been provided in the new version of the manuscript, where we added a whole Methods section on the signal-to-noise ratio (SNR) of the FSL output waveforms. We show specifically that the SNR, defined as the total comb power divided by the total ASE noise, scales as the inverse of the total spectral bandwidth, which represents an intrinsic limitation for the generation of ultra-broadband waveforms.

4. My most vehement opposition to the paper is that. There is absolutely no result in this paper which cannot just be synthesized by a commercially available RF Arbitrary waveform generator. I will not talk of synthesis of optical arbitrary waveforms since this work is not even remotely connected to that. A minimal figure of merit I expect from RF photonic techniques is that there should at least be one result which is not possible with existing standard RF equipment. This work completely fails in that regard.

Here we disagree with the referee. State-of-the-art commercial electronic AWGs have now a maximum output analog bandwidth of 45 GHz (<https://www.keysight.com/ca/en/products/arbitrary-waveform-generators/m8100-series-arbitrary-waveform-generators.html>). Electronic AWG above ~ 10 GHz is known to be extremely complex and costly, while leading to huge power consumption budgets, mainly limited by present ADC and DCA performance. Moreover, complex (amplitude and phase) optical AWG by direct temporal modulation of CW light requires the use of two synchronized electronic AWG channels. In our paper, we demonstrate optical and RF

AWG with bandwidth larger than 25 GHz and $N = 300$, using MHz-level electronics and a simple, cost-efficient photonic setup. We believe this performance by itself is already impressive. Moreover, the demonstrated operation bandwidth in our reported platform was limited by the available detection devices. By simply optimizing our current experimental set-up (i.e., compensation for the spectral un-flatness), with $f_s = 80$ MHz, and $N = 1000$, the output bandwidth could reach 80 GHz, which represents a substantial improvement as compared to presently available RF AWG techniques, while entirely avoiding the need for fast electronics. Furthermore, as discussed in the paper, the proposed concept has a solid potential to enable the synthesis of arbitrary optical waveforms with a bandwidth in the hundreds of GHz regime using presently available technologies.

Reviewers' comments:

Reviewer #1 (Remarks to the Author):

The authors made real big and fruitful efforts to address my comments and I must warmly thank them for this.

Now the principle can be intuitively seized thanks to the nice Figure 1. I sincerely like the technique, since it is simple to implement and gives results outperforming more sophisticated and costly techniques.

So I would say that I am globally satisfied by the responses to my questions and concerns.

Nevertheless, it remains some frustration when placed in the position of a possible potential user of the technique: if I want to generate a signal with a given spectral distribution (or a specific temporal waveform), I don't see how I can use the system as sketched in Figure 1 and it recalls my last comment that a gating device must be used to extract the target signal. I understand that the authors probably processed the signal by extracting a segment from the recorded scope waveform. However, from the optical point of view, if you want to generate the target optical signal, you need to temporally gate the optical output of the FSL.

The authors keep silent on the need of this gating function, even conceptually, which is neither commented nor represented graphically. It would be clearer to state very explicitly that a repetitive waveform train is generated by the system (this statement is made somewhere in the text, but nearly accidentally) and one of the sequences must be optically extracted to deliver the pure target arbitrary waveform. Figure 4a could be used to illustrate this. Please understand that the reader cannot grasp from Figure 1 that it is an AWG like it is sketched.

Finally I find moderately interesting the illustrations in Figures 3 & 4, since the time to frequency mapping is not really demonstrated in these figures (it is in Figure 2, but with a very simple signal). Why not showing less examples, but the complete process from the temporal input signal to the output waveform and spectrum?

Again I must stress about the substantial efforts made by the authors to ease the understanding by the readers and to grasp the concepts, but I think that a second pass pushing further these efforts won't be useless, for the entire benefit to the impact of the publication.

Reviewer #2 (Remarks to the Author):

In this revised manuscript, the author responds to the reviewers' questions very well, including the theoretical and experimental description sections, the noise introduced by the optical amplifier, the difference between this work and the previous work. I am very impressed by the new Figure 1. In my opinion, this improved description of the idea is much easier to understand. I still have several suggestions for the authors' reference.

1. According to Fig. 2 and its description "Moreover, as predicted, the absolute central frequency of consecutive waveforms increases as the number of roundtrips in the FSL is also increased, with a frequency increment of f_s every round-trip (τ_c), equivalent to a frequency increase of $1/\tau_c$ between consecutive individual waveforms (separated by $1/f_s$)", the center frequency difference of the Fourier transform results of the temporal waveforms at different times is related to the number of the roundtrips. It would be better to show this proportional relationship quantitatively in the manuscript.

2. As shown in Eqs. 1 and 2, " f_s is chosen such that $f_s \tau_c$ is an integer". When this integer values different, is there any difference associated with the output of the system? It would be better to have a brief discussion in the manuscript.

3. Fig. 2 lacks a, b and c in the picture.

4. Arbitrary waveform generation would not be possible if the phase is not controlled. Although in Fig. 5, the authors claim that they control the phase arbitrarily, there is no detailed method in the manuscript to realize it.

5. In the sentence of "The loop is injected with a CW laser temporally modulated by an AOM. The

latter is driven by a sine wave at $f_m = 80$ MHz, modulated in amplitude and phase by the signal under test, as detailed in the Methods section", "signal under test" is usually referred to as the signal to be measured, which might be improper to be here.

Reviewer #1 (Remarks to the Author):

The authors made real big and fruitful efforts to address my comments and I must warmly thank them for this.

Now the principle can be intuitively seized thanks to the nice Figure 1. I sincerely like the technique, since it is simple to implement and gives results outperforming more sophisticated and costly techniques.

So I would say that I am globally satisfied by the responses to my questions and concerns. Nevertheless, it remains some frustration when placed in the position of a possible potential user of the technique: if I want to generate a signal with a given spectral distribution (or a specific temporal waveform), I don't see how I can use the system as sketched in Figure 1 and it recalls my last comment that a gating device must be used to extract the target signal. I understand that the authors probably processed the signal by extracting a segment from the recorded scope waveform. However, from the optical point of view, if you want to generate the target optical signal, you need to temporally gate the optical output of the FSL.

The authors keep silent on the need of this gating function, even conceptually, which is neither commented nor represented graphically. It would be clearer to state very explicitly that a repetitive waveform train is generated by the system (this statement is made somewhere in the text, but nearly accidentally) and one of the sequences must be optically extracted to deliver the pure target arbitrary waveform. Figure 4a could be used to illustrate this. Please understand that the reader cannot grasp from Figure 1 that it is an AWG like it is sketched.

We thank the reviewer for pointing out this important issue. The reviewer is right: for AWG applications, one needs to time gate the output pulse train, to extract the intended optical (or RF) waveform. This point was not sufficiently clear from the previous manuscript version. To this aim, we have modified Fig. 1, and explicitly added a time-gating unit in the sketch of the system. We have also added a theoretical time-trace (Fig. 1, bottom), to show that the input time

signal is mapped onto the spectrum of the individual output spectrum. Finally, we have added to the text explicit descriptions of the time-gating process (e.g., see several sentences in the caption of Fig. 1, or the new sentence added in §2.2: “In practice, the optical waveform emitted at time m/f_s can be extracted from the output pulse train by time gating (e.g. by means of an intensity modulator”).

Finally I find moderately interesting the illustrations in Figures 3 & 4, since the time to frequency mapping is not really demonstrated in these figures (it is in Figure 2, but with a very simple signal). Why not showing less examples, but the complete process from the temporal input signal to the output waveform and spectrum? Again I must stress about the substantial efforts made by the authors to ease the understanding by the readers and to grasp the concepts, but I think that a second pass pushing further these efforts won't be useless, for the entire benefit to the impact of the publication.

Again, we thank the reviewer for sharing her/his impressions. To try to showcase the time to frequency mapping in our system, we have redrawn Fig. 2, and replaced the plots that give the instantaneous spectra corresponding to three different time waveforms, by a complete “waterfall” plot giving all consecutive spectra. This plot (Fig. 2.c) makes more evident the time-to-frequency mapping of the technique, by directly showing how the input flat-top signal is continuously mapped into the instantaneous output spectrum. We still would prefer to keep Fig. 3 and 4 unchanged. On the one hand, we believe that Fig. 3 provides a nice proof of the possibility of shaping the output spectrum in amplitude and in phase. On the other hand, Fig. 4 illustrates also very nicely the different steps used for OAWG, and the performance trade-offs of the proposed system.

Reviewer #2 (Remarks to the Author):

In this revised manuscript, the author responds to the reviewers' questions very well, including the theoretical and experimental description sections, the noise introduced by the optical amplifier, the difference between this work and the previous work. I am very impressed by the new Figure 1. In my opinion, this improved description of the idea is much easier to understand. I still have several suggestions for the authors' reference.

1. According to Fig. 2 and its description “Moreover, as predicted, the absolute central frequency of consecutive waveforms increases as the number of roundtrips in the FSL is also increased, with a frequency increment of f_s every round-trip (τ_c), equivalent to a frequency increase of $1/\tau_c$ between consecutive individual waveforms (separated by $1/f_s$)”, the center frequency difference of the Fourier transform results of the temporal waveforms at different times is related to the number of the roundtrips. It would be better to show this proportional relationship quantitatively in the manuscript.

We thank the reviewer for bringing up this point. As said previously, we have redrawn Fig. 2, and added a waterfall plot showing the instantaneous spectra, i.e. the power spectra of consecutive individual output traces. We believe that this change renders the time-to-frequency

mapping property more evident. We have also added a sentence to the caption of Fig. 2: "At later times, the instantaneous spectrum shifts by $1/\tau_c$ (= 9.482 MHz) every $1/f_s$ (=105.5 ns)."

2. As shown in Eqs. 1 and 2, " f_s is chosen such that $f_s \tau_c$ is an integer". When this integer values different, is there any difference associated with the output of the system? It would be better to have a brief discussion in the manuscript.

This is a good point indeed. At the end of section 2.1, we wrote the following sentence: "Finally, it is also important to note that in the case where the product $f_s \tau_c$ is not an integer, an additional quadratic component would be imprinted onto the spectral phase of the output optical waveforms, as indicated in Eq. (1), potentially providing an additional degree of control on the generated output spectra [35, 36]." We believe that this sentence is sufficient to clarify this point in the present manuscript. For more precisions, we added a reference to [36] (i.e. Guillet de Chatellus, H., Romero Cortés, L., Schnébelin, C., Burla, M., and Azaña, J., Reconfigurable photonic generation of broadband RF chirped waveforms using a single CW laser and low-frequency electronics, Nat. Commun., 9 (1), 2438 (2018).), where this property is exploited. An application of this feature could be to perform AWG by applying a modulation signal equal to the inverse fractional Fourier transform (FrFT) of the desired one. This could be interesting in specific cases, where modulating the input signal by the inverse FrFT of the desired output would be easier than doing it by its inverse FT. But we believe that this possibility lies beyond the scope of this manuscript.

3. Fig. 2 lacks a, b and c in the picture.

This point has been taken into account in the new version of Fig. 2.

4. Arbitrary waveform generation would not be possible if the phase is not controlled. Although in Fig. 5, the authors claim that they control the phase arbitrarily, there is no detailed method in the manuscript to realize it.

We thank the reviewer for raising this point. Eq. 6 shows that the phase of the output RF signal on a carrier is directly set by $\arg \left[\tilde{e}_{in} \left(\frac{f_s}{\tau_c} t \right) \right]$, i.e. by the phase of the FT of the input modulation signal. In practice, recall that to generate a given output signal (amplitude and phase), we apply to the input laser a modulation signal whose FT maps the desired profile. In our reported experiments, this operation (amplitude and phase modulation) is realized by means of an AOM modulator, where both the amplitude and the phase can be controlled. We believe the details of the experiment, particularly the AOM process used for modulating the input light wave, are clearly stated in the present manuscript. We recall for instance the following sentence (beginning of §2.2): "The latter is driven by a sine wave at $f_m = 80$ MHz, modulated in amplitude and phase by the input RF signal, as detailed in the Methods section."

5. In the sentence of "The loop is injected with a CW laser temporally modulated by an AOM. The latter is driven by a sine wave at $f_m = 80$ MHz, modulated in amplitude and phase by the signal under test, as detailed in the Methods section", "signal under test" is usually referred to as the signal to be measured, which might be improper to be here.

We thank the referee for this good remark. We have replaced “the signal under test”, with “the input RF signal.”

Hugues Guillet de Chatellus, on behalf of the authors.

REVIEWERS' COMMENTS:

Reviewer #1 (Remarks to the Author):

Thank you to the authors for addressing my comments and I consider the manuscript as suitable for publication at this stage. I wish a good success to this publication and I hope it will attract much interest in the community.

Prof. Luc Thévenaz, EPFL, Lausanne, Switzerland

Reviewer #2 (Remarks to the Author):

The authors have answered all my questions very well and I have no other questions.